# Impact of the Russian–Ukrainian Conflict on Global Food Crops

**DOI:** 10.3390/foods11192979

**Published:** 2022-09-23

**Authors:** Muh Amat Nasir, Agus Dwi Nugroho, Zoltan Lakner

**Affiliations:** 1Faculty of Agriculture, Universitas Gadjah Mada, Yogyakarta 55281, Indonesia; 2Doctoral School of Economic and Regional Sciences, Hungarian University of Agriculture and Life Sciences, 2100 Godollo, Hungary; 3Institute of Agricultural and Food Economics, Hungarian University of Agriculture and Life Sciences, 2100 Godollo, Hungary

**Keywords:** food security, price, production, trade

## Abstract

The Russian–Ukrainian conflict has been proven to cause significant losses of life and goods on both sides. This may have potentially impacted the agricultural sector. This study examines the impact of the conflict between Russia and Ukraine on the global food situation. We performed a descriptive analysis and literature review to answer this objective. Russia and Ukraine play essential roles in world food production and trade. However, the war has disrupted food production in Ukraine. Estimated Ukrainian wheat, soybean, and maize production in 2022–2023 fell precipitously. On the other hand, Russian production of these three food products shows positive growth during the same period. Furthermore, the global supply chain and food trade are hampered, causing an increase in the world’s food prices. From March to May 2022, the average global price of wheat, soybeans, and maize increased dramatically compared to during and before the COVID-19 pandemic. Finally, this poses a danger to global food security, particularly for low-income countries that depend heavily on food imports from both countries. Therefore, all countries must be prepared for the possibility that the Sustainable Development Goals cannot be achieved.

## 1. Introduction

The former USSR countries, such as Russia, Ukraine, and Kazakhstan, have only just begun to recover their grain production in the last two decades. However, several internal and external challenges, including institutional changes, land-use changes, climate change, and global economic trends, have significantly impacted their agricultural future. If these challenges can be overcome, the former USSR countries’ grain production will increase global food security [1].

Unfortunately, the former USSR countries often experience conflict. There are ideological differences between these countries. The situation has become even more complicated since NATO expanded its influence to Central and Eastern Europe. Russia is trying to preserve its hegemony in both regions through diplomacy and confrontation [2]. Likewise, Russian and Western military technologies continue to compete with one another. As a result, there is frequently competition between the two areas for influence in several countries, especially in the EU and its neighboring countries (notably Ukraine) [3].

The conflict between Russia and Ukraine in late February 2022 exacerbated the post-COVID-19 impact on the global food situation [4]. The COVID-19 pandemic is one of the causes to blame for the price increase in 2020 and 2021 [5]. The COVID-19 pandemic put the world in a new situation where, in most cases, freedom of movement was restricted and economic, cultural, and other activities, even agricultural ones, were restricted or interrupted [6]. COVID-19 has limited the mobilization of people in terms of work, including farmers. As a result, people’s incomes, welfare, and purchasing power have decreased; food prices, food insecurity, and the number of undernourished people have increased; and the global supply chain has been disrupted [7,8,9]. The food supply in many countries has fallen drastically because farmers cannot sell their products, search for market information, or look for food due to the restrictions and lockdown policies [10]. Restrictions during this pandemic have also delayed agricultural activity due to labor shortages and disrupted food distribution networks [11]. On the other hand, the COVID-19 pandemic has driven a rise in global food demand [12]. This indicates an imbalance between the demand and supply of food. As a result, during the COVID-19 pandemic, food prices and food insecurity increased globally [13].

However, the global food situation was already difficult before the COVID-19 pandemic because of civil conflicts [14], trade wars [15], climate change [16], and other factors. The trade war between the US and China was the other cause of food price spikes in 2020 and 2021, whereas food trade relations increased the efficiency and stability of food prices [17]. High import tariffs, particularly those on food products, were imposed in each country during this trade conflict [15]. Consumers must bear the cost of the increased import tariffs in the form of higher food prices.

The Russian–Ukrainian conflict has been proven to cause significant losses of life and goods on both sides [4]. This may have potentially impacted the agricultural sector. Additionally, this sector in Russia and Ukraine is highly susceptible to changes. For example, harsh weather significantly affects both countries’ agricultural productivity. In 2010/2011 and 2012/2013, grain yield was 30% below normal levels in Russia and 20% below average grades in Ukraine. In certain Russian districts, grain yield was more than 60% below average in 2010/2011 [18].

Based on the above, this study examines the impact of the conflict between Russia and Ukraine on the global food crops situation. Food crops are any plants that contain carbohydrates and proteins as human resources. This study only looked at three food crops: wheat, soybeans, and maize. The prices and presence of these three commodities on global commodity markets are highly volatile in the face of shocks [19]. In the next section, this study will discuss the impact of this conflict on food production, prices, trade, and security across the world.

## 2. Materials and Methods

This study employed annual time series data from 2020 until 2022 from various data sources, such as the United States Department of Agriculture, Food and Agriculture Organization, UN Comtrade, and Trade Map.

A descriptive or causal analysis was used to analyze the impact of the conflict between Russia and Ukraine on the global food situation. The main aspect of this analysis is that the researcher can only focus on what has happened or was happening because they have no influence over the variables. Additionally, effective research designs increase information dependability while minimizing bias [20]. Quantitative and qualitative data can be used for analysis. Descriptive studies involve quantitative data to be successful. Nevertheless, qualitative data will help in interpreting phenomena during the project investigation phase [21].

Descriptive or causal analysis can be used to understand “why” a situation has a causal effect. An excellent causal analysis can assess the effects of a situation and effectively describe population characteristics, implementation features, and the nature of the setting [22,23]. Such analysis helps the researcher in thinking systematically about many elements of a given issue, provides suggestions for more study through more analytical viewpoints and linkages, and facilitates the drawing of certain straightforward conclusions [21,24].

In addition, we conduct a literature review to provide a thorough explanation of the descriptive analysis’ findings. The literature review was carried out by reading articles published in Scopus and Web of Science.

## 3. Results

### 3.1. Global Food Production

Russia is the fourth-largest wheat producer worldwide after the EU, China, and India. At the same time, Ukraine is ranked ninth, behind the United States, Canada, Australia, and Pakistan (Figure 1).

Figure 2 shows the top 10 soybean producers worldwide, including Russia and Ukraine. By 2022, Russia was ranked eighth globally and was projected to produce 5.3 million metric tons of soybeans. Meanwhile, Ukraine was ranked 10th and was projected to produce 2.3 million metric tons of soybeans in the same year.

Russia and Ukraine are also two of the world’s primary maize producers. By 2022, Russia and Ukraine will produce 15.5 million and 19.5 million metric tons of maize, respectively (Figure 3).

Table 1 presents that during the conflict period, Russia’s average wheat and soybean production increased (2022–2023). The reverse situation occurred in Ukraine, where the production of wheat, soybeans, and maize fell drastically.

Ukraine is frequently referred to as the breadbasket of Europe. Additionally, it is a significant exporter of other essential commodities, including barley, rapeseed, and sunflower oil [26]. The Russian–Ukrainian conflict that began in March 2022 was a period of growing winter food crops (planted in late 2021 until early 2022) and sowing seeds for spring food crops (Figure 4).

### 3.2. Global Food Price

Global food prices showed an increasing trend between May 2017 and May 2022 (Figure 5). From 2017 to 2019, global food prices had a history of stability. However, food prices began to rise in 2020 and 2021. Wheat and maize prices increased each month dramatically by 2.17% and 2.59%, respectively. During the same period, the price of soybeans increased by 1.73% per month [28].

Global food prices increased drastically in 2022, especially in March, one month after the Russian attack on Ukraine. World soybean prices rose by 8.91% in March and rose again by around 0.03% and 0.46% in April and May, respectively. The price of maize increased higher than that of soybean, rising by 14.66% in March 2022, 3.77% in April 2022, and 0.95% in May 2022. Wheat saw the greatest price increase among all food products. This commodity’s price rose to 24.53% in March and then increased again in April (1.85%) and May (5.45%). According to the FAO [29], the global food and feed prices will increase by 8% to 22% beyond their current high baseline levels if this conflict is not resolved early.

The prices of wheat, soybeans, and maize were then compared across time (Table 2). During the Russian–Ukrainian crisis, the average (mean) price of the three commodities has been at its highest. Compared to normal conditions, the average price has doubled. Meanwhile, the COVID-19 period had the most price fluctuations (standard deviation). However, given that the battle between Russia and Ukraine has been going on for a short while, things could change. The probability that price fluctuations will worsen in the future is not negligible (Table 3). According to our forecast, the price of corn will skyrocket, followed by soybeans and wheat.

### 3.3. Global Food Trade and Security

The Russian Federation and Ukraine are significant food suppliers in the global market. The peak of wheat exports from both countries occurred in July and August. The Russian Federation was the top global wheat exporter, shipping 32.9 million tons of wheat and meslin (in product weight), or 18% of international shipments (Figure 6). With 20 million tons of wheat and meslin exported in 2021 and a 10% market share worldwide, Ukraine was the sixth-largest wheat exporter [29]. Wheat exports from both countries also continue to increase yearly.

The final destinations of Russian wheat exports are spread across several countries in Europe, Asia, and Africa (Figure 7). The primary destinations for Russian wheat exports are Egypt, Turkey, and Bangladesh. These three countries are also Ukrainian export destinations. Russia and Ukraine provide wheat to many Least Developed Countries (LDCs) and Low-Income Food-Deficit Countries (LIFDCs). Overall, more than 30 net wheat importers in North Africa and Western and Central Asia depend on both countries for over 30% of their wheat import needs [31]. Most interestingly, Russia and Ukraine are among Yemen’s sources of wheat. Yemen is in danger of famine because of a conflict, a drought, and a reliance on imported wheat [26,32]. In this sense, Yemen’s food security will suffer if the supply of wheat from Russia or Ukraine is disrupted.

Next, we examine the proportion of wheat exported by Russia and Ukraine to the main importers. Figure 8 shows that Russia and Ukraine are primary exporters to Bangladesh, Turkey, and Egypt. About 90% of Egypt’s wheat imports come from Russia and Ukraine. This confirms that the war between Russia and Ukraine will impact the importer countries’ food security. This is because Russia and Ukraine cannot meet the importers’ wheat needs at this time.

Our next effort was to consider the role of Russia and Ukraine as soybean exporters. Russia and Ukraine are exporters of soybeans to the world market, although in small quantities (Figure 9). Both countries’ export trends indicated a slight rise by a certain amount. The peak of Ukrainian soybean exports occurred in October, while Russian soybean exports fluctuated dramatically.

Russian soybean exports mostly go to China, Belarus, Turkey, and other countries. Likewise, exports of Ukrainian soybean to these countries also showed a significant contribution (Figure 10). However, Russia and Ukraine play a far smaller role as soybean exporters than countries such as Brazil, the US, and Argentina.

The role of Ukraine and Russia was seen to be significant in fulfilling Belarusian import demands for soybeans (Figure 11). More soybeans are imported into Belarus from Ukraine than from Russia. On the other hand, both countries’ contributions to fulfilling China’s and Turkey’s demands were less significant. Ukraine met the needs of 20.21% and 0.06%, while Russia met 2.17% and 0.69%, respectively.

Ukraine and Russia are also significant exporters of maize. The trend of maize exports from Ukraine indicates a rise in 2020 and 2021. Meanwhile, Russian maize exports mainly remained stable during that period. Ukrainian maize exports peaked at the beginning and end of the year. Meanwhile, Russian maize exports peaked in February and April (Figure 12).

The destination countries for maize exports from Ukraine are countries with upper-middle economies, such as China, the Netherlands, Egypt, Spain, and others. Likewise, Russia also sells maize to countries such as Turkey, Vietnam, the Republic of Korea, China, and others (Figure 13).

We selected China, Turkey, and Korea as samples to highlight the vital role of Ukraine and Russia in the world maize trade (Figure 14). Ukraine supplies 55.55%, 30.73%, and 12.39% of maize imports to China, Turkey, and the Republic of Korea, respectively. Meanwhile, Russia is the leading supplier of Turkey’s import needs (32.43%). Russia’s role was smaller in meeting the import needs of the Republic of Korea (2.07%) and China (1.22%). The conflict between Ukraine and Russia has disrupted the distribution of maize to China, Turkey, and the Republic of Korea and has the potential to cause food shortages in these countries.

The war has generally decreased Ukrainian exports of wheat and maize (Table 4). Ukrainian exports of wheat and maize have decreased by roughly 90% during the Russian–Ukrainian conflict, as compared to pre-Russian–Ukrainian conflict values.

## 4. Discussion

The Russian Federation and Ukraine are among the world’s most important food producers. The production of food (wheat, maize, and barley) in Russia and Ukraine could rise by up to 64% (267 million tons) by 2030 [34]. Both countries have very high food crop productivity compared to other countries [35]. This is due to fertile agricultural land, sometimes known as “black soil”, throughout Russia and Ukraine [36]. The significant role of Russia and Ukraine is further demonstrated by the fact that both countries’ fluctuating food production is a major source of uncertainty for global food markets [31].

The roles of both countries are currently disrupted due to the war. The history of human civilization demonstrates that conflict does result in significant losses in terms of the economy, society, and human life. The conflict will significantly reduce Ukrainians’ purchasing power and increase food insecurity and malnutrition. This is because people cannot engage in regular economic activity, resulting in a decreased income. The war also caused an annual average loss of 17.5% of a country’s GDP per capita [37].

According to St. Augustine’s Just War Theory, modern warfare will significantly increase the damage to human societies, their members, and the environment [38]. The conflict disrupts the production process for Ukraine because it prevents its farmers from attending to their fields, and harvesting and marketing their crops. Lang and McKee [26] reported that between 20% and 30% (around 1.9 million hectares) of areas sown for winter crops in Ukraine would remain unharvested during the 2022/23 season, and the next season’s crops cannot be planted. The places where food production and conflict were both concentrated—such as Donetsk, Kherson, and Khrakiv—saw the most significant yield losses. Other regions of Ukraine may also experience disruptions because farmers cannot find fertilizers or control pests and diseases, and are experiencing labor shortages and a lack of storage infrastructure.

Meanwhile, Russian farmers can continue their regular agricultural activities. We can see in Table 1 that Russian food production has remained stable and even increased. The agricultural sector as a whole is still expanding favorably. However, it will be challenging to market their products owing to international economic sanctions [39]. Sanctions worsen the situation rather than improve it, with substantial adverse effects on the general population’s wellbeing, including a sixfold increase in the price of food and other necessities [40]. The sanctions not only hurt Russia, but also disrupt the countries that impose them. Mardones [39] created a simulation of how sanctions will affect Russia. Economic sanctions make difficult it for Russia to export food. However, this also hurts the countries that apply sanctions, such as Latvia and Lithuania, deteriorating the condition of their food industry. Varacca and Sckokai [41] claimed that imports of food into the EU are extremely sensitive to changes in the relationship between the exporting countries and the EU. The EU will be negatively impacted if food prices in exporting countries rise. The sanctions also deal with foreign investment ceasing in Russia. At the same time, Russian agricultural investments are essential for stabilizing domestic food production and global prices [42].

According to some research, war also has increased food prices. For example, food prices have risen in Afghanistan due to the war. Since Afghanistan is culturally not the world’s primary food supplier, the impact of the price rise would only be seen domestically [43]. This could be different if the war involved the world’s food-producing countries, such as Russia and Ukraine. Food availability will decrease during a battle while demand stays the same or even increases. As a result, a gap between supply demand and food prices will grow [44].

According to Ali and Lin [45], the war increased food costs, which were transmitted to the consumer at a high price. In the food industry, the cost-efficiency effect is more than offset by the market power effect, resulting in lower food prices [46]. This has also occurred in the conflict between Russia and Ukraine, where there is an additional cost of food distribution due to increasing insurance premiums for ships planning to port in the Black Sea region [29]. This insurance is required to protect food distribution from war damage.

One month after the Russian–Ukrainian conflict began, in March 2022, world oil prices also rose by around 20.16%. Ultimately, this raises food prices [47]. This is reinforced by the fact that, under normal conditions or in the absence of war, fossil fuel prices have remained relatively stable for decades. This is the reason for the world’s low food prices [48].

Meanwhile, the conflict will undoubtedly affect the Russian and Ukrainian exports of wheat. Our opinion is based on the evidence from the Russian and Ukrainian restrictions on wheat exports in 2007/2008 [49]. As a result, the domestic markets of Russia and Ukraine are becoming less integrated with the global wheat trade. This condition causes domestic market volatility at the micro-level, increases wheat production costs, and decreases farmer incomes. Additionally, investors will scale down and postpone planned investments in Russian and Ukrainian wheat production and infrastructure [18,50]. The ongoing conflict will also undoubtedly impair the Ukrainian and Russian role as maize suppliers to several countries, especially Turkey and China. The FAO [29] stated that this would be difficult because the potential for alternative exporters is limited to entirely replacing shipments of Ukrainian and Russian maize products. For example, Argentinian exports during the current season will also likely remain limited by government efforts to control domestic inflation, while Australia has maxed its logistical shipping capacity.

The war has already resulted in the closure of Odessa, the principal Ukrainian port, and the stopping of agricultural export activities. In addition, other modes of transportation cannot make up for the Ukrainian loss of domestic marine shipping capacity, which typically handles around 90% of the countries’ commodity exports. There are also worries that the violence may harm storage and processing facilities, seaports, and inland transportation infrastructure in Ukraine [26].

Ukraine’s decrease in agricultural production and export has significantly affected the scarcity of crops and grain in their import countries, particularly in LDCs and LIFDCs [4]. Moreover, these import countries have high logistical costs, increased food loss and waste, and an uncertain quality and supply of food due to inefficient infrastructure development [51]. The FAO [29] has indicated that food insecurity will increase worldwide if the Russian–Ukrainian conflict is not resolved quickly.

Wegren et al. [52] stated that the disruption of agricultural production in Eastern Europe will cause the food security of domestic and importing countries to be threatened. For example, a country such as Eritrea depends entirely on both countries to provide for its wheat needs, with Eritrea receiving 53% of its grain from Russia and 47% from Ukraine. If Ukraine cannot export wheat due to the conflict, other countries will have to meet 47% of Eritrea’s wheat needs. The problem will be more complicated because the world food price is currently rising drastically. Meanwhile, Eritrea is included in the LDCs, meaning its proportion of food expenditure will increase, or its population will no longer be able to buy wheat.

This condition can be exacerbated for countries that depend on imports from Russia and Ukraine, as well as areas in conflict, such as Afghanistan. Afghanistan’s people are food-insecure due to the war from the early 2000s. This was brought on by a decline in food production, job losses, reduced income, and a rise in food prices [43]. Eventually, Eritrea, Afghanistan, and other countries that depend on wheat imports from Russia and Ukraine will experience food insecurity.

Even the FAO’s simulations suggest that under such a scenario, the global number of undernourished people could increase by 8 to 13 million in 2022/2023, with the most pronounced increases taking place in Asia-Pacific, followed by sub-Saharan Africa, the Near East, and North Africa [29]. This means that the world must be ready to deal with health issues brought on by food shortages in the future, such as stunting in children, poor quality of life, reduced capability for earning an income, illness, and mortality [48]. As a result, this will hinder the achievement of the SDGs and an adequate food system.

## 5. Conclusions

The conflict between Russia and Ukraine has harmed both of these countries as well as other countries. This conflict has caused the food production capacity of Ukraine to decline. Agricultural activity cannot be conducted as usual by Ukrainians. Infrastructure for the processing and distribution of food has been destroyed and rendered inoperable. Additionally, the main ports in Ukraine are blockaded. This differs from the Russian agricultural sector, which, as a whole, is still expanding favorably. Therefore, the conflict has not negatively impacted Russia, at least in terms of fundamental agricultural production. In the 2022–2023 season, Russia is also projected to export more wheat than ever before. The most pressing problem for Russia is that several countries have imposed economic sanctions, causing the country to experience barriers to exporting food products.

One month after the conflict began, in March 2022, there was a dramatic increase in the price of food throughout the world. The maize price increased by 14.66%, the soybeans price increased by 8.91%, and the wheat price increased by 24.53%. The world was concerned about this situation, mainly Russian and Ukrainian food importers, which include LDCs and LIFDCs.

This situation will threaten the achievement of the Sustainable Development Goals (SDGs), especially zero hunger. Even the FAO has estimated that 8 to 13 million more people might become undernourished globally in 2022–2023 [29], a condition that will start a domino effect that worsens the quality of health and life, increases poverty and inequality, slows economic growth, and contradicts the SDGs.

The only thing we can recommend is that all countries, notably Russia and Ukraine, use diplomacy to stop this conflict. Both Russia and Ukraine, as well as other countries, stand to gain nothing from this confrontation. Economic sanctions have also made the situation worse. Russia must stop the war and respect Ukraine’s desires. Similarly, Ukraine should prioritize strengthening relations with Russia and attempting to take a stance while avoiding the influence of other countries. We also encourage the United Nations to initiate a discussion attended by only the leaders of Russia and Ukraine to negotiate in a neutral place. Countries that support Russia and Ukraine must also intervene to bring about peace, rather than escalating the confrontation between the two. In the short term, we urge Russia to open the port in Ukraine to supply wheat, soybeans, and maize to other countries. Russia and Ukraine may involve the United Nations Peacekeeping Forces in these activities to maintain logistic transportation safety and to assuage both countries’ suspicions.

The issue of Ukrainian refugees and isolated populations must also be prioritized by UNHCR in order for them to have access to adequate food. Then, the FAO and all countries must collaborate to strengthen the food supply chain when this war is over to make it more effective and efficient. Our study’s main limitation is the scarcity of data due to the recent Russia–Ukraine conflict. As a result, we encourage future research on the impact of this conflict, including more comprehensive data.

## Figures and Tables

**Figure 1 foods-11-02979-f001:**
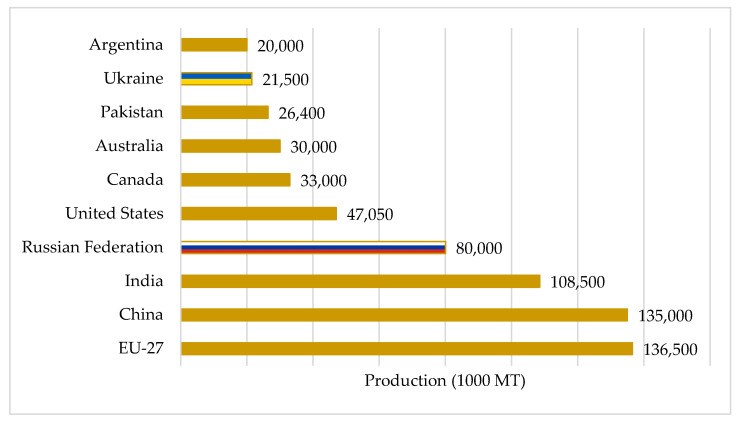
Estimated wheat production in 2022. Source: [25].

**Figure 2 foods-11-02979-f002:**
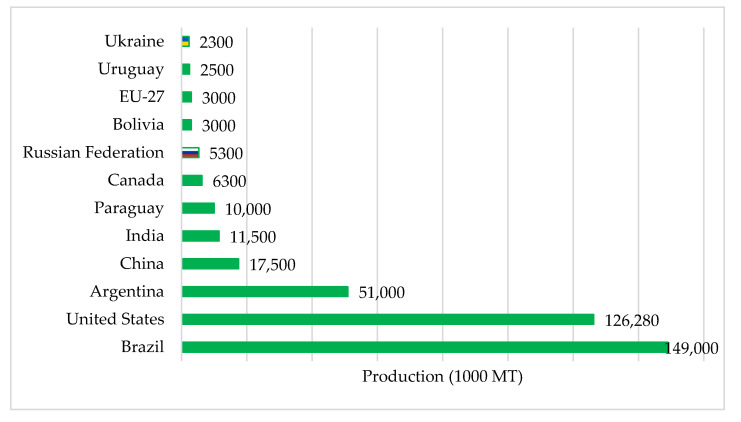
Estimated soybean production in 2022. Source: [25].

**Figure 3 foods-11-02979-f003:**
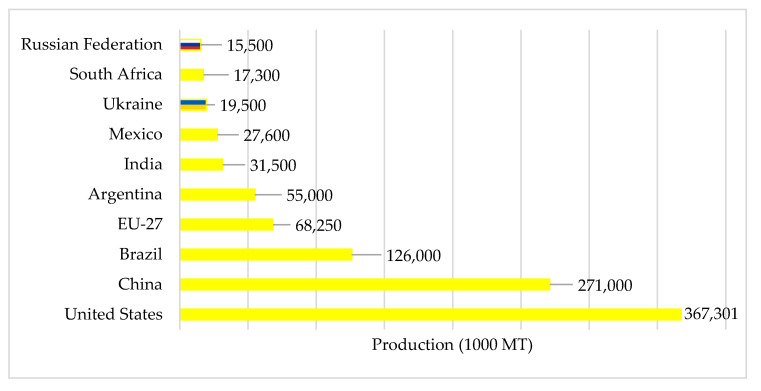
Estimated maize production in 2022. Source: [25].

**Figure 4 foods-11-02979-f004:**
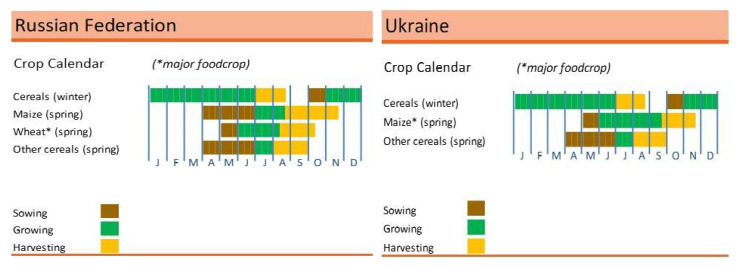
Russian and Ukrainian crop calendars. Source: [27]. ***** wheat, soybean, maize, and barley.

**Figure 5 foods-11-02979-f005:**
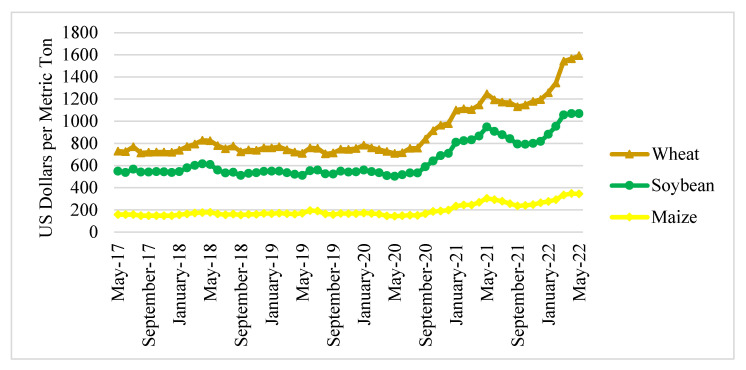
Global food prices from May 2017 to May 2022. Source: [28].

**Figure 6 foods-11-02979-f006:**
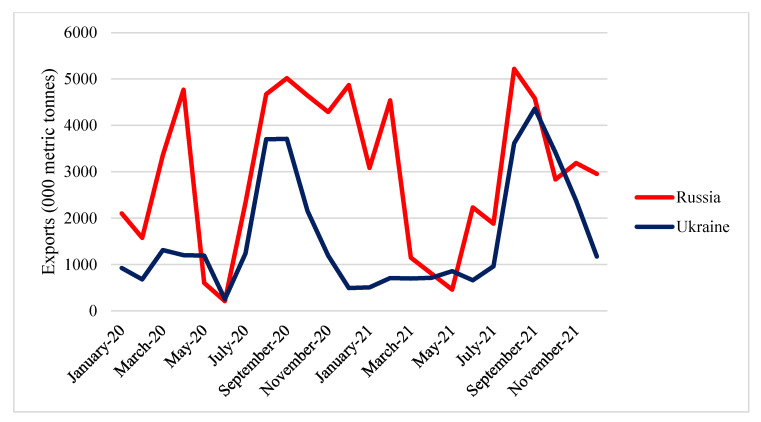
Export quantity of Russian and Ukrainian wheat in 2020 and 2021. Source: [30].

**Figure 7 foods-11-02979-f007:**
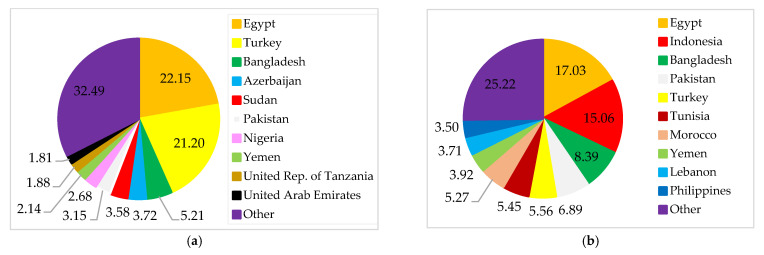
Destinations for Russian (**a**) and Ukrainian (**b**) wheat exports. Source: [30].

**Figure 8 foods-11-02979-f008:**
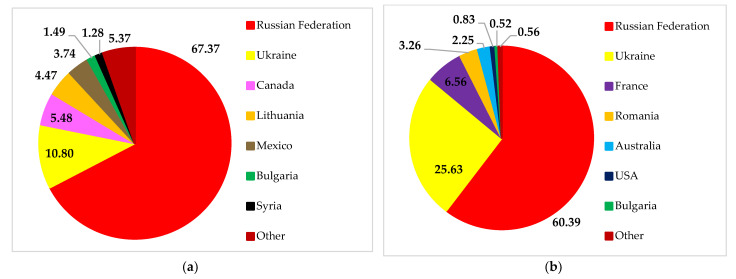
Turkey’s (**a**), Egypt’s (**b**), and Bangladesh’s (**c**) wheat import structure. Source: [30].

**Figure 9 foods-11-02979-f009:**
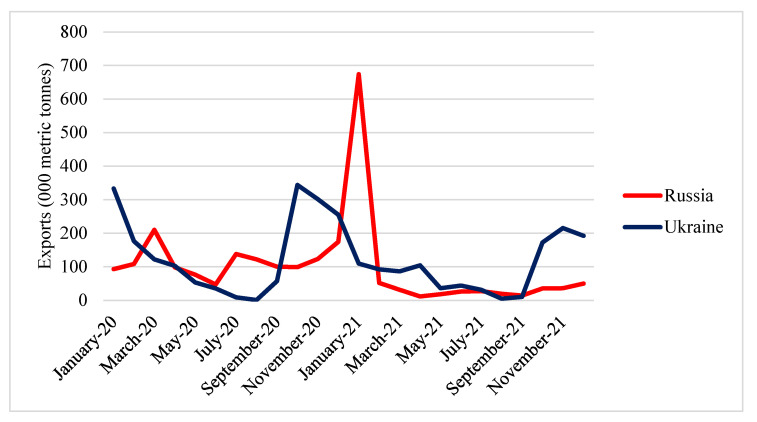
Export quantity of Russian and Ukrainian soybean in 2020 and 2021. Source: [30].

**Figure 10 foods-11-02979-f010:**
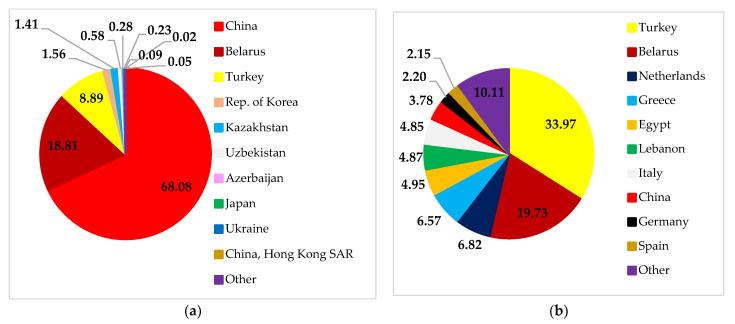
Destinations for Russian (**a**) and Ukrainian (**b**) soybean exports. Source: [30].

**Figure 11 foods-11-02979-f011:**
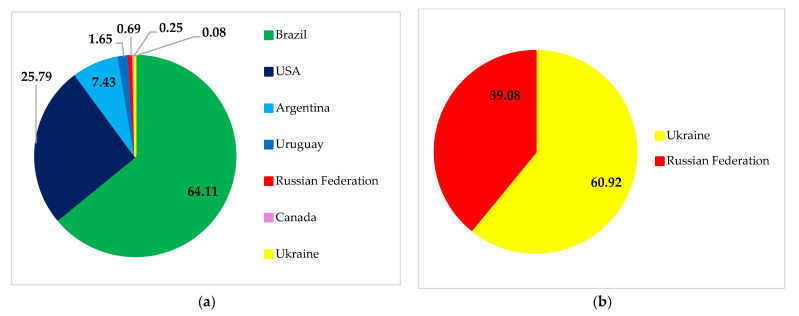
China’s (**a**), Belarus’ (**b**), and Turkey’s (**c**) soybean import structure. Source: [30].

**Figure 12 foods-11-02979-f012:**
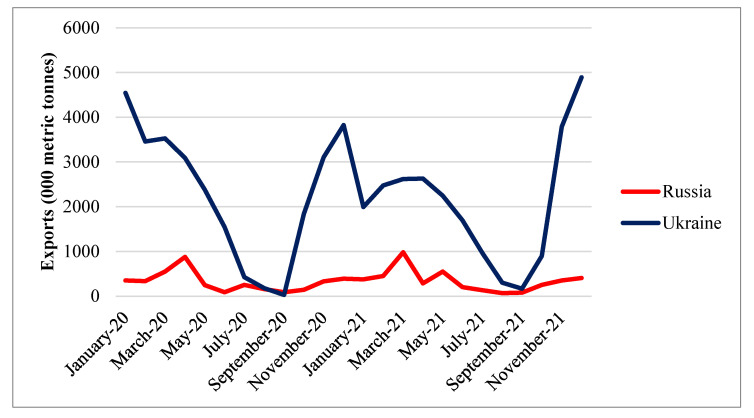
Export quantity of Russian and Ukrainian maize in 2020 and 2021. Source: [30].

**Figure 13 foods-11-02979-f013:**
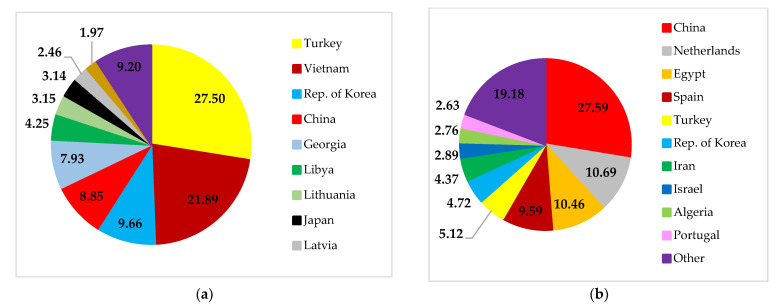
Destinations for Russian (**a**) and Ukrainian (**b**) maize exports. Source: [30].

**Figure 14 foods-11-02979-f014:**
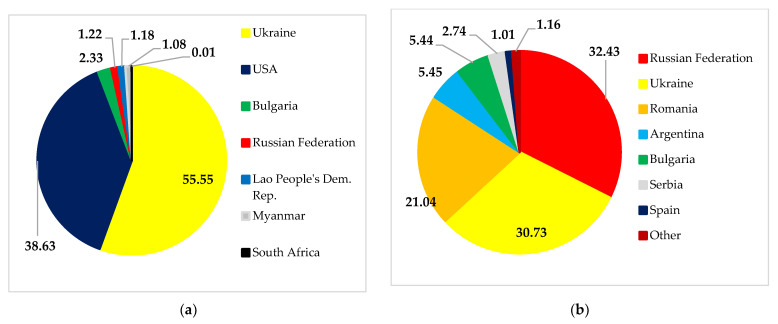
China’s (**a**), Turkey’s (**b**), and Korea’s (**c**) maize import structure. Source: [30].

**Table 1 foods-11-02979-t001:** Average production of wheat, soybeans, and maize in Russia and Ukraine in 2017–2023.

Country	Crop	2017–2021 (000 Tons)	2021–2022 (000 Tons)	2022–2023 (000 Tons)	% Change 2022–2023
Russia	Wheat	78,194	75,158	81,500	4
	Soybean	4215	4760	5300	23
	Maize	13,598	15,225	14,500	6
Ukraine	Wheat	27,927	33,007	19,500	−26
	Soybean	4023	3800	2800	−32
	Maize	33,646	42,126	25,000	−21

Source: [25].

**Table 2 foods-11-02979-t002:** Mean and standard deviation of world wheat, soybean, and maize prices (USD per metric ton).

Period		Wheat	Soybean	Maize
January 2017–December 2019 (normal)	Mean	200.04	384.27	163.44
	Std. Deviation	15.08	21.99	10.85
January 2020–February 2022 (COVID-19)	Mean	281.78	505.54	218.05
	Std. Deviation	58.16	100.81	52.77
March–May 2022	Mean	501.29	721.83	342.85
(Russian–Ukrainian conflict)	Std. Deviation	15.26	1.60	5.35

Source: [28].

**Table 3 foods-11-02979-t003:** Forecasting of world wheat, soybean, and maize prices (USD per metric ton).

Month	Wheat	Soybean	Maize
June 2022	347.8026	728.778	526.742
July 2022	350.3229	733.4661	564.842
August 2022	352.8432	738.1541	602.942
September 2022	355.3635	742.8421	641.042
October 2022	357.8838	747.5302	679.142
November 2022	360.4041	752.2182	717.242

Source: [28].

**Table 4 foods-11-02979-t004:** Ukrainian wheat and maize exports value to the world (000 USD).

Month	Wheat		Maize	
	Mean	Std. Deviation	Mean	Std. Deviation
June 2021	167,382	285,079	669,849	384,067
July 2021	92,477		282,569	
August 2021	348,050		193,356	
September 2021	710,911		104,548	
October 2021	755,401		84,788	
November 2021	632,897		266,693	
December 2021	740,857		848,268	
January 2022	101,095		1,216,216	
February 2022	87,818		874,333	
March 2022	50,587	16,822	667,186	266,819
April 2022	14,790		171,940	
May 2022	15,016		50,220	

Source: [33].

## Data Availability

The data presented in this study are available on request from the corresponding author.

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
