# Peer review of "Impact of the Russian–Ukrainian Conflict on Global Food Crops"

_foods, 2022, doi:10.3390/foods11192979_

Round 1
Reviewer 1 Report
Dear Authors
The study in hand takes up a pertinent issue in a suitable manner. Thy study provides justification for evaluating the impact of Russia-Ukrainan war globally on food crops. The methodology is good and selection of the variables is appropriate. This paper has good publishing potential but I would suggest incorporating some changes before the final consideration of this paper.
1. Abstract does not include any points related to results obtained after econometrics analysis. I think 1/2 points related to econometric results should be added in abstract.
2. In introduction part, I am unclear to find information what type of food items are considered under this paper. What author mean by food crops? Plz write it clearly
3. From line 128-132, you are discussing about price fluctuation. Support your discussion with authentic reference
4. Explain food products with reference
5. Author has mainly calculated descriptive results, In discussion part no model or theory is being explained. The theory on which whole methodology is based. So some more worth discussion should be added
Author Response
Dear Professor
Thank you so much for your efforts, suggestions, and comments. We feel that as a result of this, our manuscript will be better, systematic, and achieve high quality. We believe all of this has been done to ensure that our manuscript meets the expectations of the Foods journal.
Here is our response to your comments and suggestions
- Reviewer: Abstract does not include any points related to results obtained after econometrics analysis. I think 1/2 points related to econometric results should be added in abstract.
Author: Based on Professor's comment, we have added the results in the abstract section (line 16-18 and line 19-20)
2. Reviewer: In introduction part, I am unclear to find information what type of food items are considered under this paper. What author mean by food crops? Plz write it clearly
Author: Based on Professor's suggestion, we have added types and reasons for choosing the three food crops in the introduction section (line 71-74)
3. Reviewer: From line 128-132, you are discussing about price fluctuation. Support your discussion with authentic reference
Author: Based on Professor's comment, we have added the reference (line 141)
4. Reviewer: Explain food products with reference
Author: Based on Professor's suggestion, we have added types and reasons for choosing the three food crops in the introduction section (line 71-74)
5. Reviewer: Author has mainly calculated descriptive results, In discussion part no model or theory is being explained. The theory on which whole methodology is based. So some more worth discussion should be added
Author: Based on the Professor's suggestion, we have added St. Augustine's Just War Theory in the discussion section (line 269-270).
We hope that all of this has met your expectations.
Thank you
Best regards
Reviewer 2 Report
1. Some basic data are clear already and is it necessary to put them here? For example Figure 1 and Figure 2.
2. Figure 5 is in Chinese. Please make the correction and use the correct English.
3. Could the authors make an estimation for the coming near future based on the existing data? For example on Figure 5, Figure 6 (needs clarification on the peak), Figure 8 (no African country?), Figure 9 (clarification on the peak)
4. Discussion could be completed with the point how the transport can be done safely out from Ukraine to the third world, safely and also without carrying any dangerous materials for both countries in conflict.
5. Can you discuss a bit the possible role of the UN in this case?
6. Please put only the references that you used in the references list.

Author Response
Dear Professor
Thank you so much for your efforts, suggestions, and comments. We feel that as a result of this, our manuscript will be better, systematic, and achieve high quality. We believe all of this has been done to ensure that our manuscript meets the expectations of the Foods journal.
Here is our response to your comments and suggestions
- Reviewer: Some basic data are clear already and is it necessary to put them here? For example Figure 1 and Figure 2.
Author: Professor, we hope that these figures can give readers a better understanding, especially those who like visuals
2. Reviewer: Figure 5 is in Chinese. Please make the correction and use the correct English.
Author: Professor, we have rechecked the figures 5 and it is in English (export in US$, commodity type and time)
3. Reviewer: Could the authors make an estimation for the coming near future based on the existing data? For example on Figure 5, Figure 6 (needs clarification on the peak), Figure 8 (no African country?), Figure 9 (clarification on the peak)
Author: Professor, we have added Table 3 to estimate the coming near future. We have also added information about the peaks of food exports for the two countries in Figures 6, 9, and 12. Finally, we have also added Egypt (representative African countries) as the largest importer of wheat from Russia and Ukraine (Figure 8), but countries from Africa are not the highest importers of soybeans and corn from both countries.
4. Reviewer: Discussion could be completed with the point how the transport can be done safely out from Ukraine to the third world, safely and also without carrying any dangerous materials for both countries in conflict.
Author: Based on Professor's suggestion, we add the United Nations Peacekeeping Forces role to help food distribution in the conclusion section (line 393-394).
5. Reviewer: Can you discuss a bit the possible role of the UN in this case?
Author: Based on Professor's suggestion, we add the UN’s role to initiate a discussion between Russia and Ukraine, UNHCR to supply food for Ukrainian refugees and isolated populations, and FAO in the conclusion section (line 388-392).
6. Reviewer: Please put only the references that you used in the references list.
Author: Based on Professor's suggestion, we have corrected the reference
We hope that all of this has met your expectations.
Thank you
Best regards
Reviewer 3 Report
The article deals with the impact of the conflict in Ukraine on world cereal production. Ukraine is one of the countries with the greatest potential in this respect (especially in wheat, maize and soya). As for the current year, only forecasts are available, which are unlikely to materialize in Ukraine due to the state of war. The authors believe that the current year's grain harvest will be 20 - 30% less in Ukraine. They note that the protracted conflict will result in a reduced sowing in the autumn. The conflict is affecting the prices of basic cereals, which have doubled compared to May 2021. These are most likely self-verifying assumptions.
Based on only one source [29], the authors compare: the export directions of Ukraine and Russia grain/wheat (Figure 7), maize (Figure 10) and soybeans (Figure 13). Figures 9 and 12 compare exports of these cereals in 2020 and 2021. Circular figures 8,11 and 14, in turn, illustrate the structure of imports from a number of countries: wheat (Figure 8), maize (Figure 11) and soybeans (Figure 14), in which Ukraine and Russia have significant shares. These shares in the pie charts are shown as fragments of national flags. However, their colouring deviates significantly from reality, which I consider inappropriate. Furthermore, in Figure 8 (a,b,c) the white colour on the Russian flag blends with the colour of the paper. I also believe that in Figures 8, 10, 11 and 14 the circles should be of equal diameter. In many of them, the very small squares do not reflect the flags of Ukraine and Russia.
Author Response
Dear Professor
Thank you so much for your efforts, suggestions, and comments. We feel that as a result of this, our manuscript will be better, systematic, and achieve high quality. We believe all of this has been done to ensure that our manuscript meets the expectations of the Foods journal.
Here is our response to your comments and suggestions
1. Reviewer: The article deals with the impact of the conflict in Ukraine on world cereal production. Ukraine is one of the countries with the greatest potential in this respect (especially in wheat, maize and soya). As for the current year, only forecasts are available, which are unlikely to materialize in Ukraine due to the state of war. The authors believe that the current year's grain harvest will be 20 - 30% less in Ukraine. They note that the protracted conflict will result in a reduced sowing in the autumn. The conflict is affecting the prices of basic cereals, which have doubled compared to May 2021. These are most likely self-verifying assumptions.
Author: Professor, thank you for your comment. We tried to collect available data (although currently very limited), research results from researchers, and reports from international institutions to show the impact of this conflict. We also include Professor's comments as limitations of this manuscript in the conclusion section (line 396-398).
2. Reviewer: Based on only one source [29], the authors compare: the export directions of Ukraine and Russia grain/wheat (Figure 7), maize (Figure 10) and soybeans (Figure 13). Figures 9 and 12 compare exports of these cereals in 2020 and 2021. Circular figures 8,11 and 14, in turn, illustrate the structure of imports from a number of countries: wheat (Figure 8), maize (Figure 11) and soybeans (Figure 14), in which Ukraine and Russia have significant shares. These shares in the pie charts are shown as fragments of national flags. However, their colouring deviates significantly from reality, which I consider inappropriate. Furthermore, in Figure 8 (a,b,c) the white colour on the Russian flag blends with the colour of the paper. I also believe that in Figures 8, 10, 11 and 14 the circles should be of equal diameter. In many of them, the very small squares do not reflect the flags of Ukraine and Russia
Author: Based on Professor's suggestion, we corrected the Figures 8, 11 and 14 by replacing the flags of Russia and Ukraine with red and yellow
We hope that all of this has met your expectations.
Thank you
Best regards